# Adipose-Derived Stem Cell: “Treat or Trick”

**DOI:** 10.3390/biomedicines9111624

**Published:** 2021-11-05

**Authors:** Siti Syahira Airuddin, Ahmad Sukari Halim, Wan Azman Wan Sulaiman, Ramlah Kadir, Nur Azida Mohd Nasir

**Affiliations:** 1Reconstructive Sciences Unit, School of Medical Sciences, Health Campus, Universiti Sains Malaysia, Kubang Kerian 16150, Kelantan, Malaysia; ssyahira.work@gmail.com (S.S.A.); ashalim@usm.my (A.S.H.); wazman@usm.my (W.A.W.S.); 2Hospital Universiti Sains Malaysia, Universiti Sains Malaysia, Kubang Kerian 16150, Kelantan, Malaysia; 3Department of Immunology, School of Medical Sciences, Health Campus, Universiti Sains Malaysia, Kubang Kerian 16150, Kelantan, Malaysia; ramlahkadir@usm.my

**Keywords:** adipose tissue, adipose tissue stem cell, stem cells, cancer cells, ADSCs’ treatment, wound healing

## Abstract

Stem cells have been widely used for treating disease due to the various benefits they offer in the curing process. Several treatments using stem cells have undergone clinical trials, such as cell-based therapies for heart disease, sickle cell disease, thalassemia, etc. Adipose-derived stem cells are some of the many mesenchymal stem cells that exist in our body that can be harvested from the abdomen, thighs, etc. Adipose tissue is easy to harvest, and its stem cells can be obtained in higher volumes compared to stem cells harvested from bone marrow, for which a more invasive technique is required with a smaller volume obtained. Many scientists have expressed interest in investigating the role of adipose-derived stem cells in treating disease since their use was first described. This is due to these stem cells’ ability to differentiate into multiple lineages and secrete a variety of growth factors and proteins. Previous studies have found that the hormones, cytokines, and growth factors contained in adipose tissue play major roles in the metabolic regulation of adipose tissue, as well as in energy balance and whole-body homeostasis through their endocrine, autocrine, and paracrine functions. These are thought to be important contributors to the process of tissue repair and regeneration. However, it remains unclear how effective and safe ADSCs are in treating diseases. The research that has been carried out to date is in order to investigate the impact of ADSCs in disease treatment, as described in this review, to highlight its “trick or treat” effect in medical treatment.

## 1. Introduction

Adipose-derived stem cells (ADSCs) are mesenchymal stem cells that are known for their angiogenic properties, plasticity, and multipotent ability to differentiate into different cell lineages, such as chondrogenic, adipogenic, and osteogenic [1]. Mesenchymal stem cells can be obtained from various sites, such as bone marrow, the umbilical cord, and adipose tissue, but adipose tissue has become the optimal source for harvesting MSCs [2]. It was reported to be easier to harvest ADSCs with a high yield compared to mesenchymal stem cells harvested from bone marrow and cord blood [3]. It also reported that ADSCs have a longer life span, higher proliferative capacity, shorter doubling time, and later In-vitro senescence compared to mesenchymal stem cells derived from bone marrow [4]. Aside from that, ADSCs are multipotent and are able to differentiate into different lineages, such as neuron-like cells [5]. They can be used for the development of new blood vessels, and the induction of ADSCs into different cell types and lineages can be utilized for tissue engineering (chondrogenic, neuronal-like, osteogenic). ADSCs also contribute to the cellular turnover of adipose tissue [6]. A recent study by Hamid et al. [7] on differentiating ADSCs for chondrogenesis by co-culturing them with chondrocytes indicates that ADSCs could be used as a replacement for FBS to help grow a sufficient number of chondrogenic cells to repair cartilage injury in the near future [7]. In this review, we aimed to highlight the “trick or treat” aspect of the adipose-derived stem cells used in the treatment of diseases (Figure 1).

## 2. Part I—The Treat: Adipose Tissue Stem Cells’ Application as a Treatment

The study of cell-based treatments using adipose tissue-derived stem cells has been ongoing since Zuk et al., (2001) proposed the use of stem cells harvested from adipose tissue [8]. Their study explained the process used for the isolation of adipose-derived stem cells from lipoaspirate through a centrifugation process. The product obtained can either be applied directly as a treatment or can be further cultured and expanded in a flask for future use [8]. Since the technique was discovered, it has been used as a means for scientists to easily harvest ADSCs and use them for studies on disease treatments.

In a recent cell-based treatment study, ADSCs were used to improve bone regeneration after post-traumatic osteomyelitis due to the fact that they could differentiate into osteogenic materials. Post-traumatic osteomyelitis is a severe complication of open fractures that can be treated through surgical debridement, which is a process used to remove infected purulent bone tissue [9]. However, the surgical debridement process inhibits the bone regulation process while increasing bone resorption (osteoclastogenesis), hence affecting the healing process [9]. The process of bone resorption is mediated by an inflammatory reaction that regulates various cytokines from osteoblasts in infected bone [10]. In their study, Wagner and colleagues harvested ADSCs from the inguinal subcutaneous fat pads of C57B16 mice using an enzymatic isolation technique, before the ADSCs were transferred to bony defects right after the surgical treatment of their infected bones. The outcome of this study showed that ADSCs were able to improve bone healing via the elevation of osteoblastogenesis and downregulation of osteoclasts. In addition, stem cells were also found to downregulate the B-cell population, which is a potent activator of osteoclastogenesis activity [10].

In order to prolong rejection-free survival in an allogenic hind-limb transplantation (animal) model, an allograft was infused with ADSCs and hypoxia-primed ADSCs before being transfused via the vascular system, along with the administration of a short-term immunosuppressant treatment [11]. The result indicated that ex-vivo treatment using allografts with ADSCs prolonged the survival of the allografts, suppressed the proliferation and infiltration of T-lymphocytes, and improved the secretion of immunomodulatory cytokines, such as IL-10, through the potential induction of Treg expression in the allografts compared with the control [11].

Furthermore, the ability of adipose-derived stem cells to increase the cell proliferation rate and upregulate cells has made them an interesting material for wound-healing studies. The potential of using ADSCs as a treatment was examined through an In-vivo wound-healing assay on the application of ADSCs on circular cutaneous wounds on the back dermal skin of mice [12]. The outcomes showed that the cutaneous wound-healing rate was increased 2 to 5 weeks after the application of ADSCs [12]. ADSCs were also applied as a hair growth treatment aiming to increase the generation of new hair. Dermal papillae, which are important factors in hair regeneration, have been used for hair regeneration therapy but are difficult to isolate. Thus, dermal papillae-like tissues (DPLT) were formed by culturing ADSCs in a papilla-forming medium. In this study, wounds were created on the backs of mice; these wounds were then injected with DPLTs to evaluate the formation of new hair growth. The new hair growth throughout the treatment area showed a significantly higher number of sebaceous glands [13]. Therefore, these data suggest that engineered canine DPLTs demonstrated the characteristics of dermal papillae and had a positive effect on hair regeneration [13]. This is possible because adipose tissues are multipotent and can differentiate into other cells.

ADSCs from lipoaspirate are able to promote the outgrowth of cells [9], repopulate chondrocytes [7], repair damaged cartilage [14], and have paracrine effects on resident chondrocytes [6]. They are, therefore, good options for advancing cell-based treatment studies. Cell-based treatments using ADSCs have been applied for various purposes, including wound healing [15], the 3D bioprinting of adipose tissue for breast reconstruction [16], hair loss treatment [13,17], allograft studies [10,18], and cancer treatment. ADSCs are able to secrete a high amount of growth factors such as vascular endothelial growth factor (VEGF), hepatocyte growth factor (HGF), basic fibroblast growth factor (bFGF), platelet-derived growth factor (PDGF), keratinocyte growth factor (KGF), transforming growth factor beta 1 (TGF-β1), IGF-binding protein precursors, fibronectin, and superoxide dismutase [19]. Thus, ADSCs can have both pro-angiogenetic and anti-apoptotic effects [14,20]. Moreover, TGF-β1 secretion promotes an immunomodulatory effect and increases extra-cellular matrix deposition and collagen organization [21].

For these reasons, ADSCs have been used as an alternative medium for various cancer treatments. They have shown beneficial effects in the treatment of oral tongue squamous carcinoma (OTSCC), pancreatic cancer cells, breast cancer cells, etc. In one cancer treatment study, ADSCs were shown to have the ability to induce apoptosis and inhibit the growth of gastric cells [22]. Zhao et al. observed the growth of tumor cells by injecting ADSCs and human gastric cancer-27 (HGC-27) cells together into nude mice. After 19 days, the mice were sacrificed, and their tumors were excised and weighed. Later, a decrease in tumor formation was shown. In an In-vitro study, HGC-27 cells were cultured in ADSCs-conditioned medium (CM), showing the ability of ADSCs to secrete cell factors that would indirectly act on GC cells, causing changes in the microenvironment of HGC-27 cells. Thus, the outcome showed that ADSCs had an inhibitory effect on the proliferation of HGC-27 cells [22]. It was suggested that ADSCs could inhibit the growth of HGC-27 cells, indicating the ability of ADSCs to treat gastric cancer [22]. Observations have shown both positive and negative results depending on the type of cancer cells involved.

The rising interest in the use of ADSCs for cancer treatment can be seen through the many studies carried out aiming to observe the proliferation rate, migration rate, or influence of stem cells on cancer cell progression. The application of ADSCs in medical treatment could be improved in the future. The dual roles played by ADSCs in cancer cells, in which they are either pro-tumor or tumor inhibiting, are still being investigated in studies using various types of cancer cells. Recently, the application of ADSCs was shown to have no effect on the progression of oral tongue cancer. A study carried out by Sinha et al. indicated that ADSCs did not induce the proliferation, migration, or invasion of human tongue squamous carcinoma cells in a co-culture of ADSCs and human tongue squamous carcinoma (HSC-3) cells [23]. The authors pointed out that ADSCs do not enhance the aggressiveness of oral dysplastic and cancer cells in vitro [23]. Along with the positive results from this study, more findings can be found in Table 1, which shows the benefits of this use of ADSCs for cancer treatment.

## 3. Part II—The Trick: Side Effect of Treatment Using ADSCs

Despite the positive outcomes gained from using ADSCs in treatment, there have been some issues regarding the role of ADSCs in tumor progression. Some researchers discovered the presence of stem cell-like features in several cancers that were then identified as cancer stem cells (CSCs); it was speculated that these originated from transformed mesenchymal stem cells (MSCs) [25]. Some pre-clinical studies based on In-vitro and In-vivomodels have suggested that ADSCs may act as potential tumor promoters for different cancer cell types and support tumor progression and invasiveness through the activation of several intracellular signals [26]. Many cell-based therapy studies and observations using ADSCs have been carried out, and positive evidence has been generated, but it still remains unclear as to whether the use of ADSCs in treatment could give rise to cancer or not. In contrast, there has been much heated discussion regarding the role of adipose-derived stem cell in tumor progression. The growth factor secreted and induced by ADSCs is TGF-β, and the TGF-β/SMAD signaling pathway promotes epithelial-to-mesenchymal transition (EMT) in cancer cells [27]. In parallel, TGF-β signaling has been shown to be able to induce myofibroblastic differentiation in ADSCs exposed to breast cancer exosomes, thus promoting the desmoplastic transformation of the tumor microenvironment [28]. TGF-β is also among the main causes of the immunomodulatory effect of ADSCs. The immune-mediated response to tumors is impaired by TGF-β1, HGF, indoleamine-pyrrole 2,3-dioxygenase (IDO), and interferon gamma (IFN-γ), which are secreted by ADSCs [29]. ADSCs are also able to elicit drug resistance and cell proliferation in the breast cancer cell line MCF-7/ADR (a multidrug-resistant breast cancer cell model), mediated through the expression of C-terminal (Src) kinase (Csk)-binding protein (Cbp) [30].

The ability of ADSCs to promote tumors is mainly due to their active release of factors. In the study of Preisner et al., (2018), the team co-cultured a lower passage of primary melanoma cells and ADSCs to determine the effect of ADSCs in promoting tumors [29]. The main reason for the use of lower-passage primary melanoma cells instead of other passages is because they closely resemble cancer cells In vivo. The results obtained from this study showed an increment in HGF in melanoma cells after they were co-cultured together with ADSCs [29]. These findings showed that HGF plays an important role in affecting the growth of melanoma cells. HGF is a multifunctional cytokine that can activate various downstream pathways involved in angiogenesis, tumor growth, migration, and metastasis. Moreover, HGF has been shown to activate melanoma proliferation through MAPK signaling. This latter effect could, at least in part, be responsible for the increased proliferation of low-passage primary melanoma cell cultures. Furthermore, other regulating factors have also been detected, such as interleukin-6 (IL-6), which is responsible for proliferation, anti-apoptosis, and angiogenesis in cells [29].

There have been some reports on the impact of adipose-derived stem cells in breast cancer. There are possible interactions between primary human ADSCs and five human breast cancer cell line BRCAs (MCF-7, MDA-MB-231, SK-BR-3, ZR-75-30, and EVSA-T), pointing towards the potential increased oncological risk [31]. These results showed that ADSCs significantly affect multiple malignant features of BRCAs in vitro, such as gene expression, protein secretion, migration, and angiogenesis. Several tumor-associated proteins, such as cytokines and matrix metalloproteinases (MMPs), were found to be strongly increased in co-cultures of ADSCs and BRCAs. MMP is a protein that could possibly degrade the extracellular matrix and thereby enable invasion, as well as facilitating neo-angiogenesis and, thus, further supporting tumor growth. Altered MMP expression has already been linked to poor disease prognosis in different human cancers and enhanced cancer cell invasion [32]. In this case, ADSCs secrete high levels of various MMPs, such as MMP-1, -2, -3, -9, and -10. In addition, ADSCs promote tumor growth In-vivo by secreting interleukin-8 (IL-8), which stimulates angiogenesis and supplies nutrition, thus supporting angiogenesis [31]. A previous study found increased angiogenesis in all co-cultured BRCAs and almost all ADSC co-cultures, as well as an upregulated gene expression of IL-8 in MDA-MB-231 cell lines with ADSC co-cultures. In addition, a co-culture of ADSCs and primary BRCAs showed a strong total increase in IL-8 protein concentration, as well as a corresponding induction of angiogenesis [32].

However, in a recent study carried out by Ejaz et al., (2020), it was found that lipoaspirates and ADSCs did not increase the proliferation rate of breast cancer cells either through paracrine- or contact-dependent interactions [33]. They stated that a previous study on the effect of ADSCs in increasing tumor progression using co-culturing or an In-vivomodel method was inconsistent with the clinical scenario based on the injection of intact adipose parcels into a large tissue bed for breast reconstruction. Since most of the studies on this topic relied on In-vitro differentiated adipocytes or paracrine interactions between cells as experimental models, Ejaz et al., (2020) used a clinically relevant animal model and reported no increase in tumor size, proliferation, histological grade, or metastatic spread [33]. Even though their result contrasted with the conclusion of a previous study by Koellenspergers et al., (2017), its positive outcome might become something that scientists could build on in the future to create novel breast cancer treatments.

In short, a few genes and proteins have been discovered to be upregulated, such as those promoting angiogenesis and proneness to tumor production, but effects on the genes responsible for EMT, such as TWIST1, Snail1, Snail2, and CDH2, have not been found [32]. The upregulation of protein genes such as TNFSF-10 certainly induces apoptosis but does not kill normal cells [34]. Ejaz et al., (2020) mentioned in their study that the co-culture of ADSCs and MDA-MB-231 reduced the proliferation rate of MDA-MB-231 cell lines. Koellensperger et al., (2017) also stated that co-culturing did not significantly affect cellular proliferation [32,33]. Thus, it is possible that this was not a true reflection of the interaction of these cells with breast cancer cells, as the type of breast cancer cells used (either primary cells or lines) might influence the results. It is possible that the tested cancer cell lines might have lost some important features of cellular interactions with ADSCs throughout their process of immortalization or due to the different media conditions used for cell lines and primary cells [32]. Moreover, the level of upregulation of BRCA genes in primary breast cancer cell lines may cause differences, for example, angiogenesis factor is usually found in higher levels in primary cells than in cell lines. This is because monolayer cell lines that have not yet formed spheroids may deliver more oxygen to compact structures. Figure 2 shows the proposed signaling pathway involved in the activation of ADSCs to induce tumor progression. Meanwhile, a summary of studies showing the negative effects of ADSC treatments is provided in Table 2.

## 4. Updates and Development on Adipose-Derived Stem Cells for Clinical Applications

The current development of treatments using adipose-derived stem cells has proven their effectiveness in and benefits for curing disease. Recent updates regarding the application of ADSCs have included their use for cutaneous wound-healing treatments, as they release exosomes, which are important paracrine substances that can be used as extracellular vesicles to carry various bioactive molecules that mediate adjacent or distant intercellular communication [37]. Researchers have indicated that ADSCs-derived exosomes (ADSCs-Exos) promote skin wound healing by affecting all stages of wound healing, including regulating inflammatory response, promoting the proliferation and migration of fibroblasts and keratinocytes, facilitating angiogenesis, and regulating the remodeling of the extracellular matrix. Moreover, ADSCs-Exos are also able to improve graft retention in a manner comparable to that of their source cells, thereby achieving improved graft retention by up-regulating early inflammation [38]. The application of ADSCs-Exos has many benefits, such as their easy acquisition method, sustainable source, convenient storage and transportation, high long-term survival rate, quantitative usage, ability to avoid immune rejection, and lack of ethical problems [39]. These positive aspects show the bright side of the application of ADSCs for future stem cell-based treatments.

The emerging field of study on ADSCs has progressed beyond expectation; this research has expanded to improving the survival of ovarian tissue transplants. To overcome the problem of the massive follicle loss seen in transplanted tissue in the early post-grafting period, the benefits and properties described by ADSCs have been taken into consideration. ADSCs have been used as models of xenotransplantation, where they are embedded into a fibrin scaffold and transplanted to the peritoneum of immunodeficient mice. As expected, higher rates of the oxygenation and vascularization of ovary tissue were obtained in the early post-grafting period, along with increased follicle survival and reduced apoptosis [40]. This finding will be useful for future transplantation studies, proving that adipose stem cells can overcome the various issues if properly studied. In some cases, variations in the effectiveness of ADSCs may occur due to the differences in the method of study used, such as in vitro, In vivo, and others. It is important to determine how effective ADSCs are if administered directly to human beings. Recently, researchers tested a therapy for knee osteoarthritis based on the injection of adipose-derived stem cells to determine its efficacy and safety [41]. In this study, fat-derived cells showed a very low complication rate (16.15%), and all the complications observed were considered to be minor [41]. Additionally, ADSCs have produced promising to excellent clinical results for the treatment of knee osteoarthritis. It also appears that the use of adipose-derived stem cells is associated with clinical and radiological improvements and minimal complication rates [41].

The plastic and reconstructive surgery fields are leaders in the use of ADSCs; many practices, such as fat grafting [41,42], wound healing [43,44], and scar treatment [45], have been tested. It is overwhelming to see the growth of studies using stem cells derived from adipose tissue in other fields as well. This is promising for the health industry across the world and shows the great expansion in the knowledge of scholars and researchers. Further improvements in studies and research might lead to the discovery of many more benefits of adipose-derived tissue stem cells that could be utilized in treatments for various diseases; perhaps additional functions will be discovered. Table 3 shows some of the clinical trials that have made use of ADSCs. In the future, applications we cannot yet imagine might be discovered that will bring benefits for maintaining human health.

The stromal vascular fraction cells (SVFs) and ADSCs alone are particularly good in treating some diseases. Both are present in Stromal Vasculasr Fraction (SVF) portion, which can be obtained from human adipose tissue. Recently, the allogenic use of SVF and decellularized extracellular matrices (ECM) went through a clinical study in advancing tissue regeneration in which it could be a useful treatment alongside autologous, if the results are positive [51]. Due to the higher availability of fat, it can be easily accessed and obtained with the implementation of a fat donor program. Despite that, the safety and efficacy of the treatment are still undetermined. Based on the study mentioned, the human allogenic use of ADSCs appears to be safe and effective and surely a lot of improvement should be done in the future to determine the safest therapy. At this point, it was shown that the use of ADSCs is really good and could be exploited for various studies instead of its tricks in In-vitro and In-vivo studies.

Aside from that, the combination of ADSCs with other materials to generate or increase the proliferation of other cells is a good alternative in expanding the medical field. The characteristic of ADSC itself, which could turn into different types of cells, make it useful to be used. The right combination was possible due to the factor releases by the ADSCs, as mentioned in a previous section in this article. The same condition could be seen in the implementation of platelet-rich plasma (PRP) from the blood for tissue repair treatment due to the properties of the platelets favoring wound healing [52]. Platelets hold about 50–80 alpha-granules that contain hundreds of bioactive proteins, including a wide range of growth factors (GFs), principally represented by PDGF, FGF, VEGF, epidermal growth factor (EGF), TGF-β1, insulin-like growth factor (IGF), connecting tissue growth factor (CTGF), and HGF [53]. The In-vitro studies here reviewed state almost consistently that PRP stimulates the proliferation of the human cell [52].

The combination practice of PRP was also used together with hyaluronic acid (HA) for a wound-healing study. The study concluded that HA acts as a scaffold for the PRP [54]. The speedy development of the granulation tissue can help to shorten the healing times and reduce the need for reconstructive surgery on supplementary soft tissue. Therefore, body repair mechanisms are stimulated to heal previously irreparable tissues. In addition, the study validated that combined treatment with PRP as a bio-stimulator and HA engaged in a bio-functionalized scaffold can reduce the healing period (*p* < 0.05) and patient pain, and is cost-effective compared to traditional treatment, increasing the patient’s quality of life. Both ADSCs and PRP have shown that they could proliferate other cells [54].

In a recent update, there was positive news with the usage of stem cells as a therapy to combat with the recent COVID-19 disease. COVID-19 is the recent viral disease that could cause multi-organ damage and for which there are no particular therapies, drugs, or vaccines available for now. With the COVID-19 pandemic, stem cell therapies and especially mesenchymal stem cell (MSC)-related therapies have demonstrated their therapeutic potential for newly emerging diseases with no available treatments. So far, 88 trials were found to be registered to investigate the safety and efficacy of the transplantation therapy of stem cells or stem cell-derived exosomes for COVID-19 patients. Indications under investigation include COVID-19 with critical pneumonia, respiratory failure, ARDS, and pulmonary fibrosis [55]. Another pilot clinical study also reported on aerosol inhalation of the exosomes derived from allogenic adipose mesenchymal stem cells in the treatment of severe patients with novel coronavirus pneumonia (NCT Number: NCT04276987). In the future, we hope any great outcome will be discovered by scientists through stem cell therapy, which will help people who suffered with COVID-19.

## 5. Conclusions

In conclusion, the use of ADSCs for treating certain diseases might be effective and might be harmful. Their use could either be a treat or a trick, depending on certain factors. These factors include the type of cancer involved, as some types might benefit from the use of ADSCs, such as pancreatic cancer, breast cancer, and oral tongue squamous carcinoma. However, the use of ADSCs in colon cancer, cervical cancer, and epithelial ovarian cancer may have negative effects. At present, the positive and negative effects of ADSC(s) seem to be based on the region of the cancer, e.g., the colon, breast, or cervix, as cells in different areas react differently when being treated or co-cultured with ADSCs. In some cases, the proliferation rate of cancer cells has been promoted, while in some cases, no effect on the cells has been shown. The reason behind these inconsistent results regarding the potency of adipose-derived stem cells for the treatment of different types of cancer remains undetermined. These inconsistent results might also be because of the in vitro/In vivo/ex-vivo microenvironment itself being different to that of a human. Therefore, the results produced in the laboratory are still questionable and further exploration is still needed. The way ADSCs are administered to patients, In-vivo or in vitro, the cell growing process, the technique involved, etc., could also influence the effectiveness of ADSC(s) as a component of medical-based treatment. However, the effectiveness of ADSCs in the area of wound healing and breast augmentation/reconstruction as well as other clinical applications has been proven promising and positive by previous research carried out by several scientists. Future studies could focus on their clinical use to observe whether these cells can legitimately be regarded as a novel treatment. Further studies need to be conducted to determine whether ADSC-based treatments (sole or combination treatments) will provide a positive impact on human beings.

## Figures and Tables

**Figure 1 biomedicines-09-01624-f001:**
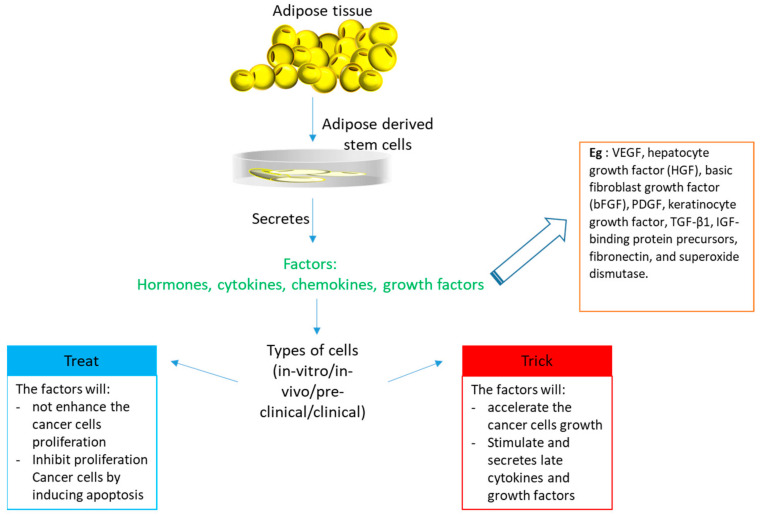
An overview of the therapeutic potential of adipose-derived stem cells for various applications.

**Figure 2 biomedicines-09-01624-f002:**
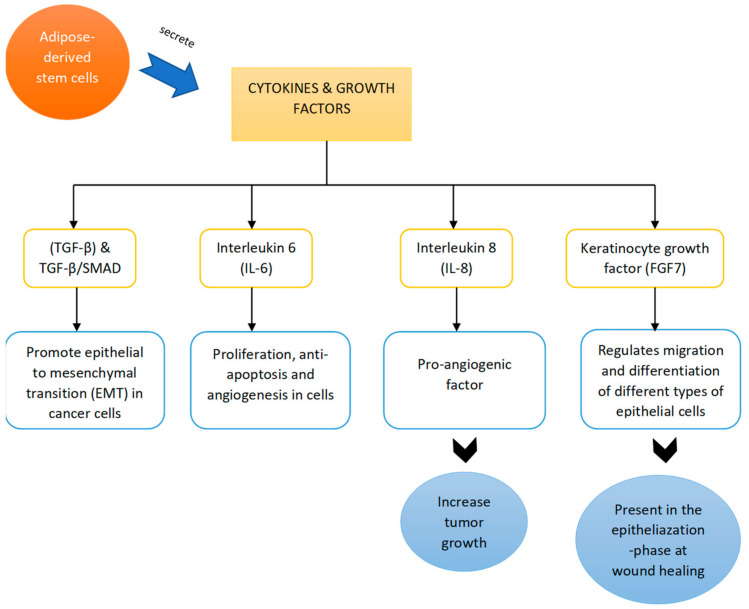
Proposed signaling pathways involved in the activation of ADSCs to induce tumor progression.

**Table 1 biomedicines-09-01624-t001:** The application of adipose-derived stem cells as a treatment for certain disease.

Source	Treatment	Type of Study	Method	Results	Ref(s)
ADSCs from subcutaneous fat pads harvested from C57Bl6 mice.	Bone regeneration after post-traumatic osteomyelitis.	In-vitro	ASCs were administered to debrided bony defects after bone infection	ASCs overcame the impairment of bone regeneration after osteomyelitis in a murine animal model.	[10]
ADSCs from lipoaspirates.	Oral tongue squamous carcinoma	In-vitro	IncuCyte wound-healing migration assay was used to study the effect of ADSCs on the cancer and dysplastic cells’ migration ability.	ADSCs did not enhance the aggressiveness of oral dysplastic and cancer cells in- vitro.	[23]
ADSC from omentum tissue	Breast cancer	In-vitro	Paracrine and contact co-culturing of breast cancer cells with ADSCs from the same donors	ADSCs did not increase the proliferation rate of the breast cancer cells either through paracrine- or contact-dependent interactions. Additionally, they inhibited MDA-MB-231 breast cancer cell contact-dependent interactions. Quantitative real-time PCR revealed no significant increase in the EMT-related genes in breast cancer cells upon co-culture with ADSCs.	[24]
Human ADSC lines	Human gastric cancer (HGC-27) cells	In-vitro and in-vivo	HGC-27 cells cultured in ADSCs-conditioned medium (in-vitro). Nude mice injected with mixture of ADSCs and HGC-27 (In-vivo)	ADSCs effectively inhibited the growth of HGC-27 cells by inducing apoptosis in-vitro and in vivo.	[22]

**Table 2 biomedicines-09-01624-t002:** The treatment using ADSCs and the negative side effect in In-vitro and In vivo.

Source	Treatment	Type of Study	Method	Results	Ref(s)
ADSCs from mouse abdominal tissue	Colon and Breast Cancer	In-vitro and in-vivo	Co-culture (in vitro) Inoculated 4T1 or CT26 cells with or without ADSCs into BALB/c mice (In vivo)	Both in-vitro and in-vivo studies showed ADSCs accelerate cancer growth. ADSC interaction with cancer cells could stimulate increased secretion of IL-6 mainly from ADSCs	[35]
ADSCs’ cell	Cervical cancer cell	In-vitro and in-vivo	Co-culture (In vitro) Injection on 6-week-old BALB/c nude mice. (In vivo)	ADSCs promoted cervical cancer growth and invasion through paracrine secretion of HGF and involvement of the HGF/c-MET signaling pathway. ADSCs secreted a high level of HGF into the supernatant	[36]
ADSC from omentum tissue	Epithelial Ovarian Cancer	*In-vitro*	Co-culture of cancel cells with ADSC	Both indirect and direct co-culture with ADSCs increased proliferation and migration ability in EOC cells to a similar extent, suggesting that the tumor-promoting effects of ADSCs are mainly mediated by the paracrine of ADSCs. MMPs released by ADSC contributed to the tumor-promoting effects of ADSCs In-vitro and In vivo.	[24]

**Table 3 biomedicines-09-01624-t003:** The treatment using ADSCs in clinical trials.

Treatments	Sample Number	Randomization/ Blinding	Source/Type of ADSCs	Application Method	Results	Ref(s)
Chronic ulcers	16 cases 24 controls Total 40	Yes/No	e-PRP from 42 cm^3^ of peripheral blood combined with ADSC from 80 mL of abdominal fat vibrated at 600 vibrations/min for 6 min and centrifuged at 52× *g* for 6 min. 5 × 10^5^ cells	Same-day procedure. 5 mL injected in multiple injections around and under the ulcer using a 10-mL syringe Follow up: 18 months	Similar healing rates. Wound-closure rates higher in case group. No adverse events	[46]
	10 cases 0 controls Total 10	No/No	Fresh, non-fractioned, non-cultured. Enzymatic congestion using collagenase and centrifugation. Donor site: abdomen 250–350 cm^3^ fat. 19.1 to 157.8 × 10^6^ cells	3–4 mL administered using a 26-gauge needle into the plane between the gastrocnemius and soleus muscles in a pattern of injections (22 per muscle, 11 in the external and 11 in the internal gastrocnemius, each one 1.5 cm to 2 cm apart) of equal volume each (0.5 mL), on either side of the midline. Follow up: 18 months	Four of six wounds closed within 9 months, one patient had a healing wound when she died at 4 months and one patient had a skin graft to close the wound at 5 months. Reduced pain in all patients. No adverse events	[47]
	10 cases 0 controls Total 10	No/No	LipoStructure^®^. Freshly purified fat using centrifugation at 3000 rpm for 3 min.	Same-day procedure. Multiple injections around and under the ulcer with 0.8-mm cannula. Follow up: 6 months	73.2% median closure rate at 3 months, 93.1% at 6 months. Reduced fibrin, necrosis, and pain. Increased granulation. No adverse events	[48]
	16 cases 0 controls Total 16	No/No	The Transpose RT™ Processing Unit (TPU) (InGeneron Inc., Houston, TX, USA) 30 mL lipoaspirate. Donor site: abdomen. 9–15 × 10^6^ cells	Same-day procedure. 4 mL injected 5 to 10 mm deep into the central and bordering ulcer area using a 1-mL Luer-Lock syringe and a 24-gauge needle. Additionally, 2.5 mL applied on a collagen sponge onto the wound. Follow up: 6 months	All venous patients and four of nine arterial-venous patients had 100% wound closure within 9–26 weeks. Reduced wound pain in all patients within days of treatment. No adverse events	[49]
Breast reconstruction Breast reconstruction	Preliminary 1 patient	No/No	Tulip low-pressure syringe lipoaspiration system was used to obtain 520 cm^3^ of lipoaspirate from the hypogastrium and the thighs.	Cell-enriched fat graft was injected into and around the defect area in multiple planes through blunt-tipped 17-gauge cannulae. As for the right breast, 50 cm^3^ of cell-enriched fat grafting were injected into the hollowness over the NAC and another 100 cm^3^ in the upper pole, while 90 and 20 cm^3^ were used in the upper and the lower pole of the left breast, respectively.	Significant contour improvement in both breasts that remained stable at 3 and 22 months of follow-up. Two minor complications occurred in the left breast; one episode of cellulitis 4 months post grafting that resolved with IV antibiotics uneventfully, and the development of a slightly painful lump 10 months post grafting that turned out to be a liponecrotic cyst after excision biopsy. Both had no impact on the cosmetic effect.	[50]
	Preliminary 1 patient	No/No	580 cm^3^ of lipoaspirate from the hypogastrium and the thighs, 260 cm^3^ of which were processed and yielded 7 cm^3^ of ASCs that subsequently enriched the remaining fat.	Skin adherence to the lateral aspect of the left breast was initially released. Overall, 320 cm^3^ of enriched fat was grafted in the periphery of the reconstructed breast in order to improve the contour and correct the tethering of the skin	No complications occurred and a satisfying cosmetic result was retained at 3 and 19 months follow-up.	[50]

## Data Availability

Not applicable.

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
