# Peer review of "Adipose-Derived Stem Cell: “Treat or Trick”"

_biomedicines, 2021, doi:10.3390/biomedicines9111624_

Round 1

Reviewer 1 Report

The review is well-written, covering many of the key aspects of the application of adipose-derived stem cells. However, it is equally important to have pictorial representations or graphics to better convey the essence of the topic of interest. So, it would be good to include a schematic showing the overview of the therapeutic potential of ASCs for various applications. Adding another schematic showing the postulated signaling pathways involved in the activation of ADSCs to induce tumor progression will also be very helpful for the readers to follow. There are several grammatical errors throughout the paper that needs to be corrected. The sentence from 44-48 lacks clarity and needs to be restructured. One of the main concerns is that the review lacks any clinical perspective since the authors fail to mention any studies which are currently under early or late-stage clinical trials. If these concerns are addressed, that will improve the quality and readability of the review paper.

Author Response

RESPONSE TO REVIEWER 1 COMMENTS

Point 1: The review is well-written, covering many of the key aspects of the application of adipose-derived stem cells. However, it is equally important to have pictorial representations or graphics to better convey the essence of the topic of interest. So, it would be good to include a schematic showing the overview of the therapeutic potential of ASCs for various applications.

Response 1: We are pleased that the reviewer found our article is well written and covering many of the key aspects of the application of adipose-derived stem cells. We also would like to thank the reviewer for reviewing our article and provided us with excellent suggestion. We agree with you that it is important to have pictorial representation or graphics to better convey the essence of the topic of interest. We now provide a figure 1 to show the overview of the therapeutic of ASCs for various application (page 2). 

Point 2: Adding another schematic showing the postulated signaling pathways involved in the activation of ADSCs to induce tumor progression will also be very helpful for the readers to follow. 

Response 2: The reviewer is correct that adding another schematic showing the postulate signaling pathways involved in the activation of ADSCs to induce tumor progression will be very helpful for the reader to follow. Therefore, we included the mentioned schematic (Figure 2) in this revised article (page 7).

Point 3: There are several grammatical errors throughout the paper that needs to be corrected.

Response 3: Reviewer raise a concern over the grammatical errors through out the paper. We already submitted this manuscript for English editing.

Point 4: The sentence from 44-48 lacks clarity and needs to be restructured

Response 4: We apologies for the unclear statement/sentences. The sentences have been re-written and restructured (page 1-2 as highlighted in yellow).

Point 4: One of the main concerns is that the review lacks any clinical perspective since the authors fail to mention any studies which are currently under early or late-stage clinical trials

Response 4: The reviewer points out an excellent point. Basically, this review wants to highlight more on the outcome of the research. There were few clinical studies included but we did not go through in details. However we did agree with the excellent suggestion, therefore we add more clinical studies in this article (page 10-12).

Reviewer 2 Report

The authors summarized the therapeutic application and the adverse effect of ADSC. However, the literature research is not comprehensive. And most the cited literatures only reported in vitro study. The expectation of this review should at least cover some finished and on-going clinical trails, and most of the pre-clinical animal studies. 

In addition, the review only covers studies related to cancer. In that case, I would strongly suggest authors to mention the research field they reviewed in title, abstract and introduction. 

There are also many grammar mistakes and typos throughout the manuscript. Please carefully proofread.

Author Response

RESPONSE TO REVIEWER 2 COMMENTS

Point 1: The expectation of this review should at least cover some finished and on-going clinical trails, and most of the pre-clinical animal studies. 

Response 1: Thank you for your review and excellent suggestions. Initially, this review wants to highlight more on the laboratory research outcomes. There were few clinical studies included but we did not go through in details. However, we did agree with the excellent suggestion, therefore we add more clinical studies in this article (page 10-12).

Point 2: The review only covers studies related to cancer. In that case, I would strongly suggest authors to mention the research field they reviewed in title, abstract and introduction.

Response 2: Thank you for pointing out this issue. As mentioned in response no.1, we already added more clinical trials in this article. Therefore, I believe there is no need to make a change in the title, abstract and introduction.

Point 3: There are also many grammar mistakes and typos throughout the manuscript. Please carefully proofread.

Response 3: Reviewer raise a concern over the grammatical errors throughout the paper. We already submitted this manuscript for English editing.

Round 2

Reviewer 1 Report

The authors have revised the manuscript based on the reviewer's suggestions.

Author Response

We would like to thank you for all you comments and suggestions

Reviewer 2 Report

I have no further questions 

Author Response

(The authors gave the same response as above.)
